# Cardiopulmonary Exercise Testing in Children and Young Adolescents after a Multisystem Inflammatory Syndrome: Physical Deconditioning or Residual Pathology?

**DOI:** 10.3390/jcm12062375

**Published:** 2023-03-19

**Authors:** Federica Gentili, Giulio Calcagni, Nicoletta Cantarutti, Emma Concetta Manno, Giulia Cafiero, Eliana Tranchita, Annamaria Salvati, Paolo Palma, Ugo Giordano, Fabrizio Drago, Attilio Turchetta

**Affiliations:** 1Complex Unit of Cardiology S. Paolo, Palidoro, Santa Marinella and Arrhythmology, Clinical Area of Fetal, Neonatal and Cardiological Sciences, Bambino Gesù Children’s Hospital, IRCCS, L.go S. Onofrio 4, 00165 Rome, Italy; 2Complex Unit of Clinical Immunology and Vaccinology, Clinical Area of University Hospital Pediatrics, Bambino Gesù Children’s Hospital, IRCCS, L.go S. Onofrio 4, 00165 Rome, Italy

**Keywords:** multisystem inflammatory syndrome in children (MIS-C), cardiopulmonary stress test (CPET), COVID-19, SARS-CoV-2, coronavirus 2, oxygen uptake, VE/VCO_2_ slope, follow-up

## Abstract

Multisystem inflammatory syndrome in children (MIS-C) is a serious health condition that imposes a long-term follow-up. The purpose of our pilot study is to evaluate the usefulness of the cardiopulmonary stress test (CPET) in the follow-up after MIS-C. All patients admitted for MIS-C in our hospital in the 12 months preceding the date of observation were considered for inclusion in the study. Pre-existing cardio-respiratory diseases and/or the lack of collaboration were the exclusion criteria. At enrolment, each subject passed a cardiological examination, rest ECG, echocardiogram, 24 h Holter-ECG, blood tests, and a CPET complete of spirometry. A total of 20 patients met the inclusion criteria (11.76 ± 3.29 years, 13 male). In contrast to the normality of all second-level investigations, CPET showed lower-than-expected peakVO_2_ and peak-oxygen-pulse values (50% of cases) and higher-than-expected VE/VCO_2-_slope values (95% of cases). A statistically significant inverse correlation was observed between P-reactive-protein values at admission and peakVO_2_/kg values (*p =* 0.034), uric acid values at admission, and peakVO_2_ (*p =* 0.011) or peak-oxygen-pulse expressed as a percentage of predicted (*p* = 0.021), NT-proBNP values at admission and peakVO_2_ expressed as a percentage of predicted (*p* = 0.046). After MIS-C (4–12 months) relevant anomalies can be observed at CPET, which can be a valuable tool in the follow-up after this condition.

## 1. Introduction

Multisystem inflammatory syndrome in children (MIS-C) is a serious health condition related to Coronavirus Disease 2019, resembling known entities such as Kawasaki disease, toxic shock syndrome and hemophagocytic lymphohistiocytosis/macrophage activation syndrome. Clinical presentation of MIS-C is dominated by significant systemic inflammation, with a multi-organ dysfunction involving cardiac, renal, respiratory, hematologic, gastrointestinal and neurological symptoms [1,2,3,4].

The multi-faceted nature of the disease course and various presentations underlines the need for prompt approach by specialists in several areas. A recent study on follow-up in MIS-C patients showed an excellent early prognosis after hospitalization and immunomodulation treatment, although few patients experienced the persistence of echocardiographic diastolic dysfunction [5].

Indeed, cardiac involvement is frequent, and some patients may have residual heart lesions, a progression of coronary involvement even after discharge and the onset or progression of arrhythmic disorders after the initial diagnosis [6]. After MIS-C, it is also possible to run into the typical disturbances of long COVID [7]. All these conditions can affect the well-being of young patients and impose the need for long-term follow-up [8].

Cardiopulmonary exercise testing (CPET) could be a very useful tool for this purpose, in support of the other methods currently used. Actually, CPET is an integrated method that allows exploration at the same time of the cardiovascular, respiratory, and muscular systems responses to a standardized effort. CPET can help to better identify the real cause of a functional limitation, or, within the same pathology, to identify which function is most affected. It can help to optimize a therapeutic intervention or to verify its effects [9,10].

To date, CPET has not been regularly used in the follow-up post-MIS-C, except in competitive athletes before resuming sports activity [11].

The purpose of our pilot study is to evaluate the usefulness of CPET as a complementary tool in the follow-up of young subjects affected by MIS-C, assessed 4–12 months after their hospital discharge.

## 2. Methods

### 2.1. Study Design

An observational study was carried out over a 12 month period and it was conducted on a small sample of young subjects who were hospitalized for MIS-C related to moderate/severe COVID-19 infection. The study conforms to the ethical principles of the Good Clinical Practice, the Helsinki Declaration, it was approved by the Ethics Committee of Bambino Gesù Children’s Hospital (protocol 2083_OPBG_2020) and it follows the current Italian regulations. Moreover, all subjects and their parents were verbally informed about the aim and the procedures of the study. All participants were assured about the anonymity of data, and that the data would be processed for scientific purposes and in an aggregate manner only.

### 2.2. Participants

All young patients (<18 years) with a history of hospitalization for MIS-C at Bambino Gesù Children’s Hospital during the 12 months prior to the observation date were considered for inclusion in the study. According to the definition given by the World Health Organization (WHO), the diagnosis of MIS-C was made on the basis of fever, laboratory markers of inflammation, multisystem (≥2) organ involvement, and temporal relationship with severe acute respiratory syndrome coronavirus 2 (SARS-CoV-2) infection [12].

Exclusion criteria for the enrolment were the presence of pre-existing cardio-respiratory diseases and/or the lack of collaboration of the young patients for CPET.

### 2.3. First-Level Evaluation

During the hospitalization, all patients had a cardiological evaluation including an electrocardiogram (ECG), echocardiogram, blood tests, chest X-ray, and abdominal ultrasound in case of intestinal symptoms (diarrhea/vomiting, abdominal pain). In selected cases, the diagnostic workup included chest computed tomography (CT) or heart magnetic resonance imaging (MRI).

At enrolment in the study, each subject passed a new cardiological examination inclusive of rest ECG, echocardiogram, 24 h Holter ECG monitoring, blood tests complete with inflammation markers, high sensitivity troponin (hs-TN), and the amino-terminal fragment of the natriuretic pro-peptide type B (NT-proBNP), as is the practice in our hospital for patients with cardiac involvement during MIS-C, in line with current recommendations [6,8]. All subjects were also questioned about the practice of physical activity before and after hospitalization, in particular on the practice of school physical activity and/or extra-curricular non-competitive or competitive sport activity.

### 2.4. Cardio-Pulmonary Exercise Test

All recruited patients were then tested with a CPET complete with spirometry. Each subject was educated on the characteristics of the test; a snack 2 h before the exam and proper hydration were recommended, and a COVID-19 swab was requested according to current hospital regulations. No subject presented contraindications to the test.

The CPET was conducted with a COSMED^®^ Quark PFT metabolimeter, regularly calibrated before each test, with breath-by-breath data acquisition, using face masks of suitable size, with continuous 12-lead electrocardiographic monitoring, finger oximetry and manual blood pressure (BP) detection with cuffs sized to fit the patient’s anthropometric characteristics. The stress test was made on a treadmill with ramped Bruce protocol [13]. Preliminarily to the test, spirometry was performed with the detection of Forced Vital Capacity (FVC), Forced Expiratory Volume in 1 Second (FEV1), and FEV1/FVC ratio. The following CPET variables were measured: peak oxygen uptake (VO_2_p) in absolute value (mL/min) and pro-kilo (mL/kg/min); oxygen uptake to heart rate ratio (oxygen pulse or VO_2_/HR) and the slope of oxygen uptake to external work ratio (VO_2_/Work Rate slope); CO_2_ production (VCO_2_) and gas ventilatory equivalents (VE/VO_2_, VE/VCO_2_); the slope of the minute ventilation (VE) to CO_2_ production ratio (VE/VCO_2_ slope), evaluated up to the ventilatory compensation point (VCP); peak minute ventilation (VE peak); breathing respiratory reserve (BRR, expressed as the difference in litres between MVV and VE peak). The anaerobic threshold (AT) was estimated by V slope method and ventilator equivalent method [14], and oxygen uptake at AT was recorded. The VCP was identified where VE started to change out of proportion to VCO_2_ with the increase of VE/VCO_2_ and the consequent decline of the CO_2_ end tidal pressure (PETCO_2_) [15,16]. All CPET results were compared to Burstein’s paediatric reference values [17]. The VE/VCO_2_ slope values were also compared with other paediatric predicted values, such as Dilbert’s [18], Takken [19], and Blunchard’s [20].

This study was approved by the local ethics committee (protocol 2083_OPBG_2020) and was in accordance with the current version of the Declaration of Helsinki (2013). Patients and guardians signed informed consent to the study.

### 2.5. Statistical Analysis

All statistical analysis was performed by SPSS Statistics 21 (IBM Corporation, Armonk, NY, USA). Categorical variables are expressed as absolute numbers or percentages. Continuous variables are presented as mean value and standard deviation (SD).

A correlation (Pearson’s correlation) and a logistic regression analysis were researched between the CPET parameters, and some indices of severity of the disease, such as the length of hospitalization, the need for combined therapy, the NT-proBNP, and TnT values. Moreover, a logistic regression analysis was performed to correlate changes in inflammatory markers with changes in CPET parameters. All variables with significant results at univariable analysis were entered into multivariable analysis. A *p*-value was considered significant when ≤0.05.

## 3. Results

From a total population of 36 children hospitalized for MIS-C in our hospital, 20 patients (55.6%) met the inclusion criteria. Table 1 shows the main demographic and anthropometric characteristics of our cohort.

All subjects at the time of admission had heart involvement with 8 cases (40%) of myopericarditis without frank reduction of the left ventricular ejection fraction (LVEF), 10 (50%) with frank reduction of LVEF, 2 (10%) with Kawasaki-like coronary involvement. The diagnosis of myopericarditis was made on the basis of the elevation of myocardionecrosis enzymes associated with echographic and/or MRI findings compatible with acute inflammatory alterations.

A total of 11 out of 20 patients (55%) presented interstitial pneumonia on X-ray and/or chest CT.

A total of 18 out of 20 patients (90%) with gastrointestinal symptoms (diarrhea, vomiting, abdominal pain) showed a bowel inflammatory involvement associated with signs of intestinal inflammation on abdominal ultrasound. An increase in hs-TN and in NT-proBNP was observed in 16 patients (80%), and 1 patient (5%) showed an isolated increase in NT-proBNP associated with hepatic involvement and peritoneal effusion.

During the hospitalization, all subjects required immunoglobulin therapy and steroids, 4 (20%) needed heart inotropes (adrenaline and milrinone), 3 (15%) respiratory support, and one (5%) required continuous renal replacement therapy (CRRT) purification treatment with Cytosorb filter in order to reduce excessive levels of inflammation mediators.

The length of hospitalization ranged from 11 to 43 days (19.24 ± 8.47).

Subjects were enrolled 4 to 12 months after discharge (mean 9.75 months ± 3.23). At enrolment, all subjects reported having had a very sedentary lifestyle since discharge, free from school physical activity, which was regularly practiced by all subjects before admission; only 7 of the 20 subjects practiced non-competitive extra-curricular sport activity before hospital admission, but not regularly in the last two years due to the restrictions related to the pandemic; no one practiced competitive activity.

At enrolment, all subjects were declared to be asymptomatic, and there were no significant abnormalities in the blood chemistry tests. In particular, the inflammation markers values were normal, there were no echocardiographic abnormalities (4 subjects with mild mitral valve insufficiency, 1 with mild dilation of the aortic root with minimal aortic insufficiency), and with regard to the analysis of the Holter ECG, none of the subjects investigated documented the presence of supraventricular or ventricular extrabeats in numbers greater than 50 beats/24 h, and no complex arrhythmias were observed. All the subjects showed sinusal rhythm, a normal mean HR (between 60–90 bpm), normal AV and IV conduction, and none showed significative pause >2.5 s. The basal spirometry was unremarkable. Non-specific ventricular repolarization anomalies (VRAs) were observed in three patients, however, there was no previous ECG to compare their presence. Table 2 shows the main clinical instrumental characteristics of our population at admission and at the enrolment. 

In contrast to the substantial normality of all the second-level investigations, some parameters of the CPET were altered (Table 3), in particular:-A total of 7 out of 20 subjects (35%) showed a slight (less than 80% but more than 70% of the predicted) reduction in VO_2_p values compared to the expected [17], while 3 cases (15%) showed a moderate VO_2_p reduction with VO_2_p values below 70% of the expected (Figure 1). Among them, two children showed a major cardiological involvement with the need for ventilatory support and inotropic therapy;-The mean value of the oxygen pulse was 9.0 ± 3.0 mL/beat; similarly to VO_2_p, its results were lower than 80% of predicted [17] in 10 cases and in 3 subjects lower than 70% of predicted;-A total of 15 out of 20 cases (75%) showed VO_2_ at AT values lower than the 60% of predicted [17];-The mean value of the VE/VCO_2_ slope was 33.4, with 19 of the 20 children recruited in the study (95%) showing higher VE/VCO_2_ slope values than the Burstein predicted values [17]. Similar results were observed using different prediction formulas that can be adopted in healthy paediatric populations [18,19,20] Figure 2.

At univariable analysis, a statistically significant inverse correlation was observed between the following:-CRP values at admission and peak VO_2_/kg values (*p =* 0.034);-Uric acid values at admission and peak VO_2_ expressed as a percentage of predicted (*p* = 0.011);-Uric acid values at admission and peak oxygen pulse expressed as a percentage of predicted (*p* = 0.021);-NT-proBNP values at admission and peak VO_2_ expressed as a percentage of predicted (*p* = 0.046).

These correlations were not confirmed on multivariate analysis. No significant Pearson’s correlations were found between the CPET parameters and some indices of severity of the disease, such as the length of hospitalization, the need for combined therapy, and the NT-proBNP and TnT values.

## 4. Discussion

MIS-C is characterized by systemic inflammation with a wide variety of signs and symptoms, according to the most affected systems [1,2,3,4]. The long-term outcomes are not well known yet and will emerge over longer follow-ups, just as the therapeutic approach will have to evolve in relation to a better knowledge of the pathogenic mechanisms of the disease [2].

To date, the primary outcomes identified are the worst left ventricular ejection fraction, the highest coronary artery z-score (of the left anterior descending or right coronary artery) [21], the presence of long-term myocardial scars at MRI [22], the occurrence of non-cardiac organ dysfunction, inflammation, and major medical events [21].

After MIS-C, it is also possible to run into the typical disturbances of long COVID [7]. In the study of Messiah et al., 27% of children with MIS-C examined reported chronic long COVID symptoms, with the three most reported symptoms being tiredness, headache, and difficulty thinking [7].

All these conditions can affect the well-being of young patients and their return to daily school activity and home routines and require careful monitoring. To date, the follow-up of these patients has not included the use of CPET, except for those who intend to get back into competitive sports in some countries [11].

The CPET is an exam that, in addition to assessing the functional capacity of a subject, permits the simultaneous study of the response of the cardiovascular, respiratory, and muscular systems to a standardized effort, allowing better identification of the real cause of the functional limitation and to optimize any therapeutic intervention [9,10]. For these reasons, we believe that the CPET could be particularly useful in the long-term follow-up of young patients after MIS-C.

In our pilot study, at 4 to 12 months after hospitalization for COVID-19-related MIS-C, the only significantly altered test among those performed in the paediatric population recruited for the study was the CPET. Alterations in CPET parameters, such as lower VO_2_p and higher VE/VCO_2_ slope values than predicted, were recently described in a series report involving 5 patients who survived MIS-C, without pre-existing paediatric chronic conditions and evaluated 1.3–6.2 months after discharge, although in absence of echocardiographic abnormalities in 4 of them [23].

In our study, 7 patients (75%) showed VO_2_p values lower than 80% of predicted, and 3 (30%) of them were even lower than 70% of what was expected.

The low VO_2_p values observed in our population can be partly explained by the important muscle deconditioning of these young patients (unfortunately not investigated at the time of the analysis) and the poor habit of movement of all subjects under examination. None of them, indeed, had yet resumed regular physical activity after the initial suggestion to refrain from sports, in line with current recommendations [6,8].

It is known that VO_2_ is a parameter related to the training status of the subject [9]. A failure to increase or a decrease in peak VO_2_p [24], or more generally a reduction in exercise tolerance [25] have been widely described in the juvenile population in association with COVID-19 confinement.

The low VO_2_ at AT values (15 out of 20) observed, as well as the low oxygen pulse values (less than 80% of the predicted in 10 out of 20 subjects, although with a normal trajectory of the curve in all subjects) seem to support this hypothesis.

Nevertheless, before attributing low values of VO_2_p and oxygen pulse to sedentary lifestyle or muscle deconditioning alone, caution is necessary. Frankly, low values of VO_2_p could hide residual cardiovascular and respiratory distress, and it is known that VO_2_p is an independent risk factor associated with poor prognosis in several diseases [10,26,27] and all-cause mortality in the general population [28].

In our cohort, two of the three children with the lowest VO_2_p values had had major cardiological involvement with the need for ventilatory support and inotropic therapy, and a statistically significant correlation was observed between NT-proBNP levels at admission and peak VO_2_ values (*p =* 0.046).

The statistically significant inverse correlation observed between CRP values at entry and peak VO_2_/kg values (*p* = 0.034), likewise, could be interpreted as a slower recovery of cardiopulmonary parameters in subjects with greater initial inflammatory involvement.

Similarly, the significant inverse correlation between uric acid values and VO_2_p (*p* = 0.011) or peak pulsed oxygen (*p =* 0.021) values at CPET, could suggest that subjects with a greater degree of initial impairment show slower cardiopulmonary recovery. In this regard, it is known that hyperuricemia with or without deposition has important systemic cardiovascular, nephrological, and metabolic implications related mainly to endothelial damage and inflammation, and should be evaluated in the Covid era [29,30].

The lack of other significant correlations in our study between CPET values and some other disease-severity indices (length of hospitalization, the need for combined therapy, TnT values, alterations in the blood count, D-dimero, firbinogen, etc.) could be partly related to the substantial homogeneity of the clinical picture of the subjects analysed, as well as to the small number of the sample observed. Further investigation will be necessary, in our opinion, in the young patients with the lowest VO_2_p values.

Another relevant piece of data emerged from our study on this small paediatric population after hospitalization for MIS-C following moderate/severe COVID-19: we observed a higher slope in VE/VCO_2_ ratio with respect to expected values. The exercise VE relative to VCO_2_ is an important parameter that shows complementary information about ventilatory limitation and ventilatory control [31,32].

The VE/VCO_2_ slope response to exercise is indicative of disease severity as well as prognosis in patients with chronic heart failure, pulmonary hypertension, interstitial lung disease, and chronic obstructive pulmonary disease [33].

Although no significant correlations were observed in our cohort between VE/VCO_2_ slope values and some indices of disease severity (length of hospitalization, need for combination therapy, TnT values, CBC alterations, CRP, D-dimers, fibrinogen, etc.), we believe that the low numerosity and substantial homogeneity of the population under observation make definitive conclusions difficult.

Some authors documented how, in adult subjects who recovered from hospitalized COVID-19, without baseline confounders, more than one-fourth (29%) had high values of VE/VCO_2_ slope being responsible for an exercise ventilatory inefficiency [34] after more than five months (169 days ± 28 days) from subjects’ discharge.

Similar findings emerged from the already quoted study by Astley et al. involving five post-MIS-C patients, where high VE/VCO_2_ slope values were reported in all subjects [23].

It is known that in children aged 10 years or below, there is a higher VE/VCO_2_ slope, both for a higher exercise respiratory rate and for a lower partial pressure of carbon dioxide (PaCO_2_) set point [35], with a progressive decrease during the second decade of life [36].

In addition, there is still no homogeneity in the literature on the reference values for slope VE/VCO_2_ in children, even if the prediction formulas proposed by Burstein et al. seem to be quite reliable, as they come from an adequately large sample of subjects [17]. Anyway, comparing what was recorded in this small population with different paediatric expected values obtained on a cycloergometer [17,18,19] and treadmill [20], a definitely high percentage of subjects showed elevated values of VE/VCO_2_ slope (Figure 2). Worthy of note is the fact that the mean value of the VE/VCO_2_ slope found in our study was 33.4, higher than the average value reported by Gavotto (mean value of 29) [37] and Rhodes (mean value < 28) [38] in the respective control groups evaluated with CPET in their studies on congenital heart disease in children.

Again, the high VE/VCO_2_ slope values observed could be in part related to the state of complete physical detraining of these children, leading to an excessive ventilatory response to work [9]. This could be corroborated by the observed reduced VO_2_p values in many of them, and by the fact that this parameter can get better with physical training [39,40,41]. A possible alternative might lie in an altered central control of ventilation [42,43,44,45] or in an underlying vascular dysfunction, described in post-MIS-C children [23].

Nonetheless, high VE/VCO_2_ slope values seem an interesting finding that deserves further research in young subjects who suffered from MIS-C after moderate/severe COVID-19.

It would be useful, in addition, to re-evaluate over time how these parameters change, particularly in relation to the resumption of physical activity.

## 5. Limitations

One of the limits of our study is the small size of our sample. However, this is a pilot study, and MIS-C is an uncommon condition after COVID-19 in children. In addition, our inclusion criteria were quite strict considering younger subjects without previous diseases.

Another limitation is the lack of a control healthy group even if we used common prediction formulas to evaluate the CPET performance in our subjects.

Moreover, 9 subjects of our series did not reach a fully maximal test (RQ < 1.1) even if RQ values between 1.05 and 1.09 are considered maximal in paediatric age if associated with a plateau in the HR and/or VO_2_ curve [46], conditions present in our young patients.

Furthermore, the lack of a group of subjects who had moderate/severe COVID-19 infection without MIS-C does not allow us to define whether the observed CPET alterations have to be related to COVID-19 alone or to COVID-19 that triggered MIS-C.

Finally, it is not possible to exclude that the wide time span from the acute phase of the MIS-C to the ergometric evaluation may have influenced the results. Nonetheless, in our opinion, the value of the data found remains significant. The longitudinal evaluation of the same subjects, currently underway, could provide further information in this sense.

## 6. Conclusions

This pilot study, conducted on a small sample of young subjects who were hospitalized for MIS-C related to moderate/severe COVID-19 infection, shows that even in the absence of significant anomalies in the second-level cardiorespiratory tests, relevant anomalies can be observed at CPET. In particular, reduced exercise tolerance and a high slope of the VE/VCO_2_ ratio were observed in most of our patients, with a statistically significant correlation between some hematochemical markers of disease severity and some parameters at CPET.

A poor habit of movement with important muscle deconditioning related to long hospitalization and the period of convalescence from MIS-C could be the main causes of such alterations, even if other causes need to be further investigated.

CPET can be a valuable and non-invasive tool to quantify the functional limitation of these patients and monitor their progresses over time, adding physiological aspects that can be really useful in the follow-up of these patients.

## Figures and Tables

**Figure 1 jcm-12-02375-f001:**
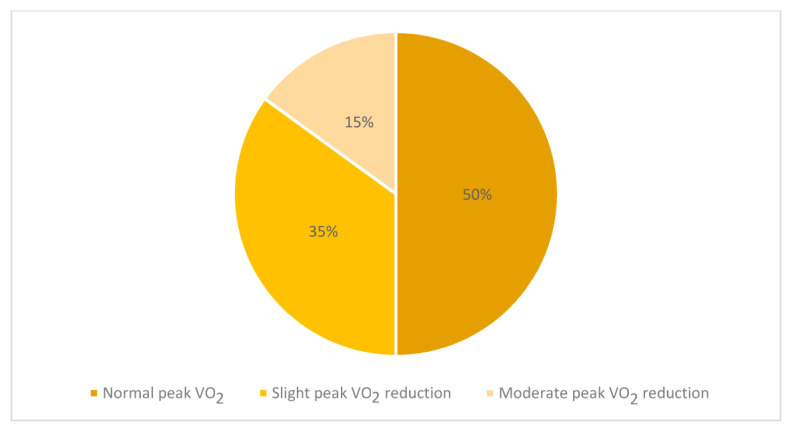
Percentage of cases with normal, slightly reduced, and moderately reduced VO_2_p.

**Figure 2 jcm-12-02375-f002:**
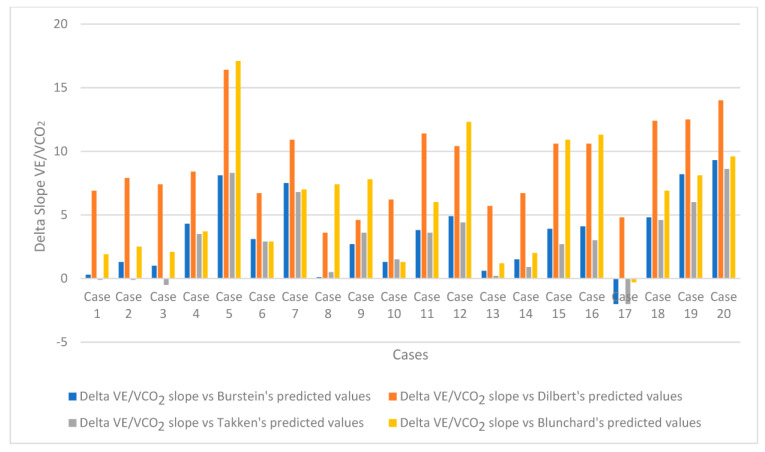
Delta VE/VCO_2_ slope versus different predicted values.

**Table 1 jcm-12-02375-t001:** Main demographic and anthropometric characteristics of our cohort at cardiopulmonary exercise test. Data are expressed as mean (and standard deviation) or as count (and percentage). Legend: BMI = body mass index.

Demographic and Anthropometric Characteristics	
Age (years)	11.76 ± 3.29
Gender	
Male	13 (65%)
Female	7 (35%)
Weight (kg)	50.91 ± 17.67
Height (cm)	149.20 ± 14.67
BMI (kg/m^2^)	22.22 ± 4.76

**Table 2 jcm-12-02375-t002:** The main clinical-instrumental and laboratory findings data at the time of admission and at enrolment.

	At Admission	At Enrolment
Symptoms	20 (100%)	0 (0%)
School physical activity	20 (100%)	0 (0%)
Extra-curricular non-competitive sport activity	7 (35%)	0 (0%)
High Inflammatory markers	20 (100%)	0 (0%)
Heart involvement	20 (100%)	0 (0%) ^
R-VLEF	10 (50.0%)	0 (0%)
Signs of myo-pericarditis ^^	8 (40.0%)	0 (0%)
Kawasaki-like alterations ^^^	2 (10.0%)	0 (0%)
Lung involvement *	11 (55%)	0 (0%)
Bowel involvement **	13 (65%)	0 (0%)
Need of immunoglobulin therapy	20 (100%)	0 (0%)
Need of steroids	20 (100%)	0 (0%)
Need of inotropic support	4 (23.5%)	0 (0%)
Need of respiratory support	3 (17.6%)	0 (0%)
Need of CRRT purification treatment	1 (5.6%)	0 (0%)
ECG -and Holter ECG anomalies		
Complex or frequent arrhythmias	0 (100%)	0 (0%)
VRAs	18 (90%)	3 (15%)
AVB II or III	0 (0%)	0 (%)
hs-TnT (pg/mL)	148.47 (±363.54) (increased in 16 patients, 80%)	<14 ng/mL °
NT-proBNP (pg/mL)	4434.78 (±6233.53) (increased in 17 patients, 85%)	<317 ng/mL °°
Hemoglobin (g/dL)	11.5 ± 1.7	13.7 ± 1
Leukocytes (10^3^/µL)	9.9 ± 3.4	6.6 ± 1.9
Lymphocytes (10^3^/µL)	1.2 ± 0.6	2.5 ± 0.8
Lymphocytes (%)	12.7 ± 8.8	38.9 ± 8.1
Platelets (10^3^/µL)	258.1 ± 173.7	261.9 ± 80.1
Serum albumin (g/dL)	3.2 ± 0.6	4.7 ± 0.3
Ferritin (ng/mL)	862.1 ± 717.5	52.7 ± 26.4
Triglycerides (mg/dL)	173.8 ± 72.1	87.9 ± 47.4
Uric acid (mg/dL)	3.8 ± 1.4	4.5 ± 1.4
Serum sodium (mEq/L)	133.3 ± 4.5	139.9 ± 0.8
C-reactive protein (mg/dL)	13.4 ± 8	0.1 ± 0.1
D-dimers (µg/mL FEU)	2.8 ± 4.2	0.3 ± 0.0
Fibrinogen (mg/dL)	616.8 ± 193.6	306.6 ± 62.6

Clinical description at the time of admission and at CPET. Data are expressed as mean (and standard deviation) or as count (and percentage) when appropriate. ^ A total of 4 (20%) subjects with mild mitral valve insufficiency, 1 (5%) with mild dilation of the aortic root with minimal aortic insufficiency. ^^ Echo images compatible with acute inflammatory alterations (pericardial effusion, pericardial hyperechogenicity, comets, impaired left ventricular contraction). ^^^ Iperechogenic coronary artery with focal dilation. * Signs of interstitial pneumonia on X-ray and/or chest CT. ** Gastrointestinal symptoms (diarrhea, vomiting, abdominal pain) associated with signs of intestinal inflammation on abdominal ultrasound. Legend: R-LVEF = reduced left ventricular ejection fraction (R-LVEF). VRAs = ventricular repolarization abnormalities. AVB II or III = second- or third-degree atrioventricular block; CRRT = continuous renal replacement therapy. Laboratory findings at the time of admission and at CPET. Data are expressed as mean (and standard deviation) or as count (and percentage) when appropriate. ° hs-TnT normal value < 14 ng/mL; °° NT-pro-BNP normal value < 317 ng/mL. hs-TnT = high sensitivity troponin; NT-proBNP = aminoterminal fragment of the natriuretic pro-peptide type B.

**Table 3 jcm-12-02375-t003:** Detailed summary of the CPET main results.

Basal HR (bpm)	83.75 ± 12.9
Peak HR (bpm)	192.35 ± 7.59
Peak HR (% of expected)	92.39 ± 4.05
Peak RER	1.12 ± 0.04
Peak systolic blood pressure	140 ± 18
Peak diastolic blood pressure	69 ± 8
Basal systolic blood pressure	104 ± 13
Basal diastolic blood pressure	62 ± 9
Peak VO_2_ (mL/min)	1676.20 ± 521.03 (lower than expected in 10 out of 20)
Peak VO_2_/kg (mL/min/kg)	34.16 ± 6.99 (lower than expected in 10 out of 20)
Peak VO_2_ (% of expected)	84.95 ± 16.53 (lower than expected in 10 out of 20)
VO_2_ at AT (ml/min)	1157 ± 383
VO_2_ at AT/Kg (ml/min/kg)	23 ± 4
VO_2_ at AT (% of predicted VO_2_max)	55 ± 10 (lower than 60% of predicted VO_2_max in 15 out of 20)
Oxygen Pulse (mL/beat)	9.0 ± 3.0
Oxygen Pulse (% of expected)	85 ± 16 (lower than 80% of predicted in 10 out of 20 and lower than 70% in 3 out 20)
Slope VE/VCO_2_ (VCP)	33.40 ± 3.54
Slope delta Burstein ^1^	3.45 ± 3.08 (higher than expected in 19 out of 20)
Slope delta Dilbert ^2^	8.84 ± 3.37 (higher than expected in 16 out of 20)
Delta Takken ^3^	2.92 ± 2.95 (higher than expected in 16 out of 20)
Delta Blunchard ^4^	6.43 ± 4.97 (higher than expected in 19 out of 20)

Cardiopulmonary exercise testing results in 20 patients post MIS-C. Data are expressed as mean (and standard deviation) or as count (and percentage) when appropriate. ^1^ Mean difference ± standard deviation between VE/VCO_2_ slope values and the predicted according to Burstein. ^2^ Mean difference ± standard deviation between VE/VCO_2_ slope values and the predicted according to Dilbert. ^3^ Mean difference ± standard deviation between VE/VCO_2_ slope values and the predicted according to Takken. ^4^ Mean difference ± standard deviation between VE/VCO_2_ slope values and the predicted according to Blunchard. Legend: HR = heart rate; RER = respiratory exchange ratio; VO_2_ = oxygen uptake; VE = external ventilation; VCO_2_ = carbon dioxide production; VCP = ventilatory compensation point.

## Data Availability

The data that support the findings of this study are stored in Hospital database and are available from the corresponding author on a reasonable request.

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
