# Peer review of "Cardiopulmonary Exercise Testing in Children and Young Adolescents after a Multisystem Inflammatory Syndrome: Physical Deconditioning or Residual Pathology?"

_jcm, 2023, doi:10.3390/jcm12062375_

Round 1
Reviewer 1 Report
Dear authors
Thank you for your selected scientific work. I shouldn't comment on the approaches, but you should improve some. I would probably expect the most (not the monitoring of classic blood tests) new approaches and monitoring of cardiological markers, as these parameters are already being determined, especially in pre-operative blocks. Here they could be used because it was a pilot study and use several options and outline directions that will be used not only by the authors, but also by the wider scientific community. Since it is also necessary to check whether the authors do not copy their previous results, data, I checked the publications of the first author for the last two years, and some works came to me more appropriately edited and more consistently written than this one. That's why I leave free space for personal editing.
Also for the outlined reason, I ask for a complete blood analysis to be completed (and to be careful about various other markers that are forgotten, such as uric acid, for example). Even with defining which parameters were set. It is not obvious from your description.
Secondly, I repeat this to almost every team, but when you deal with the cardio-topic of action, effects and changes, it is necessary to mention cardiovascular parameters such as BP (SBP, DBP), HR and others (you know you had a holter, the least is to download and evaluate the data ) that were recorded. It is the basic characteristic of the file. Even if it is a case file.
As for your results, um .... what statistics did you use? Because somehow I don't see where you used it, so they probably didn't use it. Simply, the article itself contains only descriptive statistics, which are more suitable for philosophizing.
The description of the methodological part is poor and clustered. Couldn't it be described more professionally?
On the other hand, I like the use of CPET and think that it could bring many interesting new contributions to basic research. A big plus.
Lines 227-229 I had the feeling that it was written in the work that the young individuals did not move anyway, as they should have resumed physical activity (nothing bad, but that is their primary problem, since in our country it is said that in a healthy body, healthy spirit). At this point, I just thought, why wasn't the control of individuals who had an active lifestyle done?
Line 245-246 I totally agree
And then another question comes to mind: with Covid 19, there was a breathing disorder associated with pulmonary hypertension. Were angiotensins and ACE markers monitored??
Rare 282 -283 are you continuing your research with these young probands? Were they instructed to exercise for a period of time and re-track the VE/VCO2 ratio?
Line 290 it would be appropriate to communicate and reach out to other institutions in other countries. The sample would immediately be larger and more diverse, which is more beneficial in terms of the significance of some statistical parameters.
Line 315: I don't understand the reason why a different font is used.
Reviewer 2 Report
Dear authors,
With interest I have read the manuscript entitled “Cardiopulmonary exercise testing in children and young adolescence after a multisystem inflammatory syndrome: physical deconditioning or residual pathology?” The manuscript is interesting and would greatly add to the literature. However, some points should be clarified before it is considered for publication. Please see my comments below:
Abstract:
The authors stated that “relevant changes can be observed at CPET”. My understanding is that there was no baseline CPET for the authors to evaluate changes, and this phrase can be interpreted like that. Please rephrase it clarifying this point.
Introduction
The introduction is clear and well written. On the purpose of the study, consider to indicate if the objective would be having the CPET as an isolated or complementary tool to monitor individuals with MIS-C.
Methods
Third paragraph: is the ECG, echocardiogram, 24-hour Holter ECG monitoring, and blood pressure completed after 4-12 months part of the hospital clinical routine to perform after the hospital discharge of individuals with MIS-C? Is enrolment referring to enrollment in the study? Please clarify
Firth paragraph: Did the participants receive any specific recommendations in advance to their test? Were any contra-indication considered? If so, did you have any cases in which the contra-indications applied?
The timeline of 4-12 months is a large range. Do the authors believe that the fact that you have used a range of time, instead of a specific time point could have influenced the results?
Results
The authors found no correlation between CPET parameters and some indices of severity of the disease such as length of hospitalization, the need of combined therapy, the NT-proBNP and TnT values. Do the authors think that the fact that the CPET was not conducted on the moment of the hospital discharge could have influenced these results? I am considering here that individuals were restoring their functional capacity during the period of time considered in the study (4-12 months).
Discussion
Paragraphs three, four and five are to me, the justification of the study and should go under the introduction section.
Line 261: How long after the hospital discharge were these individuals evaluated?
The lack of a CPET right after the hospital discharge could also be considered as a limitation in my point of view. On a monitoring aspect, it made it difficult to understand how compromised the cardiorespiratory fitness of the included individuals were right after their COVID experience, and how it progressed (or was restored) over time.
Round 2
Reviewer 2 Report
Dear Authors,
Thank you for submitting the revised version of your manuscript. The quality of the text has been improved and I have no further suggestions. Congratulations on your work,